# A FREQUENCY PERSPECTIVE OF ADVERSARIAL ROBUSTNESS

## ABSTRACT

Adversarial examples pose a unique challenge for deep learning systems. Despite recent advances in both attacks and defenses, there is still a lack of clarity and consensus in the community about the true nature and underlying properties of adversarial examples. A deep understanding of these examples can provide new insights towards the development of more effective attacks and defenses. Driven by the common misconception that adversarial examples are high-frequency noise, we present a frequency-based understanding of adversarial examples, supported by theoretical and empirical findings. Our analysis shows that adversarial examples are neither in high-frequency nor in low-frequency components, but are simply dataset dependent. Particularly, we highlight the glaring disparities between models trained on CIFAR-10 and ImageNet-derived datasets. Utilizing this framework, we analyze many intriguing properties of training robust models with frequency constraints, and propose a frequency-based explanation for the commonly observed *accuracy vs robustness* trade-off.

## 1 INTRODUCTION AND BACKGROUND

Since the introduction of adversarial examples by Szegedy et al. (2014), there has been a curiosity in the community around the nature and mechanisms of adversarial vulnerability. There exists an ever-growing body of work focused on attacking neural networks starting with the simple FGSM (Goodfellow et al., 2015), followed by the advanced PGD (Madry et al., 2018), a stronger C&W attack (Carlini & Wagner, 2016), the sparser Deep Fool (Su et al., 2019) and recently even a parameter free Auto-Attack (Croce & Hein, 2020). These methods and algorithms are consistently countered by the adversarial defense community, starting with distillation-based methods (Papernot et al., 2016), logit-based approaches (Kannan et al., 2018), then moving on to the simple, yet powerful PGD training (Madry et al., 2018), ensemble-based methods (Tramèr et al., 2018) and various other schemes (Zhang et al., 2019; Xie et al., 2019). Despite the immense progress made by the field, there exist many unanswered questions and ambiguities regarding these methods and adversarial examples themselves. Several works (Athalye et al., 2018; Kolter & Wong, 2018; Croce & Hein, 2020; Carlini & Wagner, 2017) have raised doubts about the efficacy of many methods and have made appeals to the research community to be more vigilant and skeptical with new defenses.

Meanwhile, there exists a thriving research corpus dedicated to deeply studying and understanding adversarial examples themselves. Ilyas et al. (2019) presented a feature-based analysis of adversarial examples, while Jere et al. (2019) presented preliminary work on PCA-based analysis of adversarial examples, which was followed up with Jere et al. (2020) offering a nuanced view of the same through the lens of SVD. Ortiz-Jimenez et al. (2020) derive insights from the margins of classifiers.

Given the intriguing nature of adversarial examples, another way of examining them is through the signal processing perspective of frequencies. Tsuzuku & Sato (2019) first proposed a frequency framework by studying the sensitivity of CNN's for different Fourier bases. Yin et al. (2019) then pursued a related direction where they explored the frequency properties of neural networks with respect to additive noise. Abello et al. (2021) explore how the frequency properties of the image itself affect the model's outputs and robustness. Caro et al. (2021) studied whether convolution operations themselves have an intrinsic frequency bias. Guo et al. (2019) came up with the first variant of adversarial attacks which target the low frequencies and Sharma et al. (2019) strengthened this line of thought by showing that such attacks had a high success rate against adversarially defended

models. Deng & Karam (2020) proposed a method of generating adversarial attacks in the frequency domain itself. Complementary to these, there have been efforts by Lorenz et al. (2021) and Wang et al. (2020a) in detecting or mitigating adversarial examples by training in the frequency domain.

These works also analyzed the nature of adversarial examples under the purview of frequencies and tried to arrive at an explanation for their nature. Wang et al. (2020b) hypothesized how CNNs exploit high frequency components, leading to less robust models, which is also the primary argument for a class of pre-processing based defenses, e.g., those based on JPEG. Wang et al. (2020d) also had arguments in support of this conjecture, based on their analysis on CIFAR-10 (Krizhevsky, 2009). It is confounding that these results are at odds with the successful low frequency adversarial attacks by Sharma et al. (2019) and raises the pertinent question: *What is the true nature of adversarial examples in the frequency domain?* Our work challenges some pre-existing notions about the nature of adversarial examples in the frequency domain and arrives at a more nuanced understanding that is well rooted in theory and backed by extensive empirical observations spanning multiple datasets. Some of our observations overlap with insights from the concurrent work by Bernhard et al. (2021) and offers additional evidence in this ongoing debate. Based on these, we arrive at a new framework that explains many properties of adversarial examples, through the lens of frequency analysis. We also carry out the first detailed analysis on the behaviour of frequency-constrained adversarial training. Our key contributions can be summarized as follows:

- We show that adversarial examples are neither high frequency nor low frequency phenomena. It is more nuanced than this dichotomous explanation.
- We propose variations of adversarial training by coupling it with frequency-space analysis, leading us to some intriguing properties of adversarial examples.
- We propose a new framework of frequency-based robustness analysis that also helps explain and control the accuracy vs robustness trade-off during adversarial training.

The rest of the paper is organized as follows: we first start off with basic notations and preliminaries. Then we introduce our main findings about adversarial examples in frequency domain and subsequently present a detailed analysis about their properties, complemented by extensive experiments.

## 2 PRELIMINARIES

We denote a neural network with parameter $\theta$ by $y = h(x; \theta)$, which takes in an input image $x \in \mathbb{R}^{H \times W}$ (omitting the channel dimension for brevity) and outputs $y \in \mathbb{R}^C$ where $C$ is the number of classes. Let $D$ and $D^{-1}$ represent the forward Type-II DCT (Discrete Cosine Transform) (Ahmed et al., 1974) and its corresponding inverse. The DCT breaks down the input signal and expresses it as a linear combination of cosine basis functions. Its inverse recovers the input signal from this representation. For a 1-D signal, the $k^{\text{th}}$-freq of $x \in \mathbb{R}^N$ and its corresponding inverse is given by

$$D(x)[k] = g[k] = \sum_{n=0}^{N-1} x_n \lambda_k \cos \frac{(2n+1)k\pi}{2N}, \tag{1}$$

$$D^{-1}(x) = x[n] = \sum_{k=0}^{N-1} g[k] \lambda_k \cos \frac{(2n+1)k\pi}{2N}, \tag{2}$$

$$\text{where } k = \{0, 1, \dots, N-1\} \text{ and } \lambda_k = \begin{cases} \sqrt{\frac{1}{N}} \text{ for } k = 0 \\ \sqrt{\frac{2}{N}} \text{ else.} \end{cases} \tag{3}$$

We denote an adversarial attack that is bound by budget $\epsilon$ by

$$\max_{||\delta||_p \leq \epsilon} \mathcal{L}(h(x + \delta; \theta), y) \tag{4}$$

where $\mathcal{L}$ is the loss associated with the network and $\delta$ is the adversarial noise bounded under a defined $L_p$ norm to be less than perturbation budget $\epsilon$. We perform a standard PGD-style update (Madry et al., 2018) to solve this maximization problem via gradient ascent and for an attack bounded by an $L_p$ norm and step size $\alpha$, the adversarial noise is given by

$$\delta = \arg\max_{||V||_p \leq \alpha} V^T \nabla_x \mathcal{L}(h(x; \theta), y) \tag{5}$$

where $V$ is the direction of steepest normalized descent. Now, to generate an adversarial example that consists of certain frequencies, we restrict its adversarial noise $\delta$ to a subspace $S$ defined by $S = \text{Span}\{f_1, f_2, \ldots, f_k\}$, where $f_i$ are orthogonal DCT modes and $k \leq N$,

$$\delta_f = \underset{||V||_p \leq \alpha}{\arg\max} V^T D^{-1}(D(\nabla_x \mathcal{L}(h(x;\theta), y)) \odot M) \tag{6}$$

$$\text{where } M_z(X) = \begin{cases} 1 \text{ if } D(X_z) \in S \\ 0 \text{ if } D(X_z) \notin S \end{cases} \quad \text{is the mask to select frequencies.} \tag{7}$$

In our work, we consider the $L_\infty$ and $L_2$ norms, solving for which gives us the update steps:

$$\delta_f = \alpha \cdot \text{Sgn}(D^{-1}(D(\nabla_x \mathcal{L} \odot M))) \text{ for } L_\infty \text{ and} \tag{8}$$

$$\delta_f = \alpha \cdot D^{-1}\left(D\left(\frac{\nabla_x \mathcal{L} \odot M}{||\nabla_x \mathcal{L} \odot M||_2}\right)\right) \text{ for } L_2 \tag{9}$$

We refer to this method as *DCT-PGD* in the rest of the paper. Note that the manual step size selection of standard PGD is not always accurate, leading to discrepancies in robustness measures as illustrated in Lorenz et al. (2021). Hence, we provide our results and observations with a DCT version of Auto-Attack. Unless mentioned otherwise, we utilize the ResNet-18 architecture for all models in our experiments. We use the term *adversarial training* to refer to the method by Madry et al. (2018) for all models, with the exception for ImageNet models where we use Adversarial training for free method (Shafahi et al., 2019). We utilize $L_\infty$ norm with $\epsilon$ of 4/255 for TinyImageNet and ImageNet datasets and $\epsilon$ of 8/255 for CIFAR-10 in all our experiments. Exact training details are included in the Appendix A.2. The terms *low frequencies* refer to frequency bands 0 to 32 and *high frequencies* refer to frequency bands bands 33 to 63.

## 3  WHY DO WE NEED A FREQUENCY PERSPECTIVE?

The focus of the community has been mostly on generating adversarial examples which are *indistinguishable* to humans, but can easily fool models. This notion gave rise to the incorrect assumption that since these perturbations are *imperceptible* to humans and they generally have to be in higher frequencies. The assumption was solidified when various pre-processing defense methods like Gaussian blur and JPEG showed initial success, further adding to confirmation bias. The fallacy is a classic case of *Post Hoc Ergo Propter Hoc*, i.e., the outcome of events is influenced by the mere ordering. Most of these experiments were centered only around CIFAR-10 and one can easily observe that the efficacy of such methods are questionable when extended to larger datasets like ImageNet and TinyImageNet (e.g., see (Dziugaite et al., 2016; Das et al., 2017; Xu et al., 2018)). This incorrect assumption has also led to claims about adversarial training *shifting the importance of frequencies* from the higher to the lower end of the spectrum  (Wang et al., 2020c;b). As we show in the subsequent sections, this is not entirely true.

We contend that this entire framework of investigating adversarial examples (e.g., blocking high frequency components using Gaussian blur pre-processing) is flawed, as one cannot verify the converse setting of blocking low frequency components. This is because low frequency components are inherently tied with labels (Wang et al., 2020b), conflating the two phenomena. Contrary to these, we argue and show that *adversarial examples are neither high frequency nor low frequency and are dependent on the dataset.*

## 4  NATURE OF ADVERSARIAL SAMPLES IN FREQUENCY SPACE

### 4.1  NOISE GRADIENTS

Measuring the change of output with respect to the input is a fundamental aspect of system design. Whether it is a controls circuit or a mathematical model, the measure $\frac{dy}{dx}$ gives us valuable information about the working of the model. When the model in question is a black box, like a neural network, the measure is invaluable as often it is our only insight into the inner mechanisms of the model. In the case of a classifier, the measure $\frac{dy}{dx}$ is a tensor that is the same size as the input, which tells us about the impact of each pixel in input $x$ on the resulting output $y$. Drucker & Le Cun (1991)

first applied this concept on neural networks and called them *input gradients*. Over the recent years, this measure and its variants have found a new home in the model interpretability community (Selvaraju et al., 2016; Wang et al., 2019), where it forms the bedrock for various improvements.

Taking a cue from this, we propose to measure $\frac{dy}{d\delta}$ or ***Noise Gradients***, which inform us about the regions of noise, which have maximal impact on the output $y$. In our work, we are more interested in the frequency properties of adversarial examples, and hence take this one step further and propose to measure the **DCT of noise gradients**, i.e., $D\left(\frac{dy}{d\delta}\right)$ or $D\left(\nabla_\delta Y\right)$. In a sense, we are measuring the model's *reaction* to different frequency components in the adversarial input. This tensor $D\left(\nabla_\delta Y\right)$ (which has same shape as input) will point us towards the specific frequencies that affect the output y of the model. To analyze the adversarial frequency properties of a given dataset, we calculate the *average noise gradients* with respect to the model, under both normal training and adversarial training paradigms. Once computed, it will paint a picture about the interplay of adversarial noise and frequencies.

### 4.1.1 ANALYSIS OF NOISE GRADIENTS

We define the quantity $D\left(\nabla_\delta Y\right)_f$ as the noise gradient at frequency $f$. Note that this quantity is useful because it differs from $D(\delta)$ by at most a constant multiple, i.e.,

$$D(\delta) \propto D\left(\nabla_\delta Y\right) \tag{10}$$

*Proof.* Let $\hat{x} = x + \delta$ where $\delta$ is the adversarial noise, then

$$\nabla_\delta Y = \nabla_x Y = \nabla_{\hat{x}} Y \quad \text{and} \quad \nabla_\delta L = \nabla_x L = \nabla_{\hat{x}} L \tag{11}$$

We have (from Appendix A.1),

$$\nabla_x L \propto \nabla_x Y \tag{12}$$

From the definition of the PGD update step, we have

$$\delta = \alpha \cdot \nabla_x L = \alpha \cdot \nabla_\delta L \tag{13}$$

for some constant $\alpha$. Taking DCT of both sides, and by the linearity of the DCT, we have

$$D(\delta) = D(\alpha \cdot \nabla_\delta L) = \alpha \cdot D(\nabla_\delta L), \text{ and therefore, } D(\delta) \propto D(\nabla_\delta L). \tag{14}$$

Now from equation 11 and equation 12 we get

$$D(\nabla_\delta L) \propto D(\nabla_\delta Y) \tag{15}$$

Using this in equation 14, we have

$$D(\delta) \propto D(\nabla_\delta Y) \tag{16}$$

$\square$

We see that the term $D(\nabla_\delta Y)$ corresponds to the frequencies that are affected by adversarial noise.

### 4.1.2 EMPIRICAL OBSERVATION OF NOISE GRADIENTS

We compute the average DCT of noise gradients over validation sets of TinyImageNet, CIFAR-10, and ImageNet datasets for models with normal and adversarial training under attack from a PGD-based $L_\infty$ adversary. The resulting tensors are visualized in Figure 1a. It shows the path taken by the PGD attack in the frequency domain under different scenarios for different datasets. We see that for normally trained CIFAR-10 models, the DCT of noise gradient activations are towards the higher frequencies and they gradually shift towards lower frequencies once the model is adversarially trained. Whereas for TinyImageNet and ImageNet models, we observe that the activations are already in lower-mid frequencies and adversarial training further concentrates them. These results clearly establish the following:

- The DCT content of PGD attacks is highly dataset-dependent and we cannot make general arguments regarding frequency nature of adversarial samples just based on training.

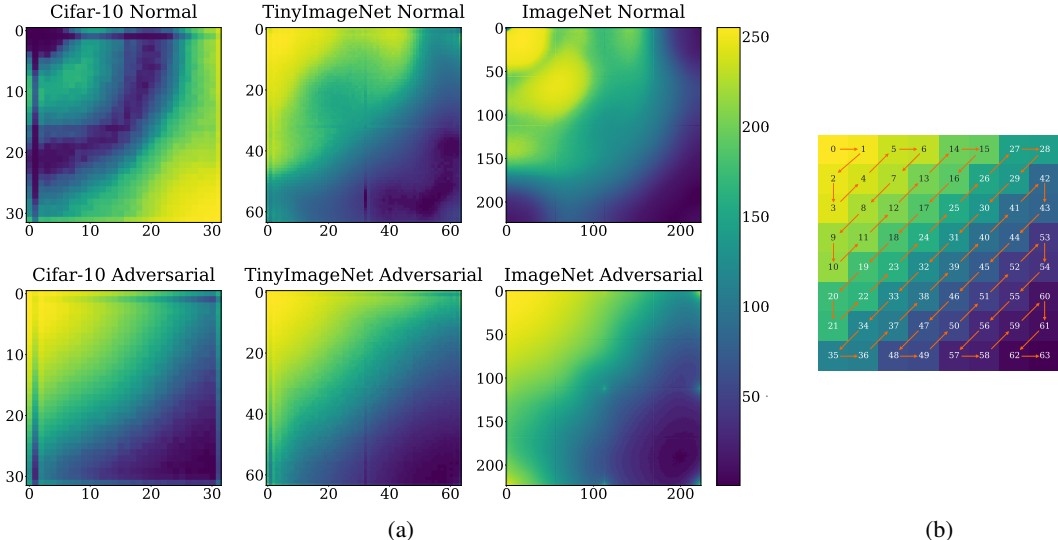

Figure 1: (a) The **DCT of Noise Gradients** averaged across the validation sets, visualized with histogram equalization. (b) shows the standard 8×8 DCT block with the all 64 frequencies arranged in zigzag order.

- The notion that adversarial training *shifts* the model focus from higher to lower frequencies is not entirely true. In many datasets, the model is already biased towards the lower end of the spectrum even before adversarial training.

- To verify that this phenomenon is attributed to the dataset alone, we also observe similar behaviour across other architectures (Appendix A.5), across different image sizes (Appendix A.6) and for different attacks like $L_2$ (Appendix A.7).

## 5 MEASURING IMPORTANCE OF FREQUENCY COMPONENTS

To examine the properties and behaviour of adversarial examples in the frequency domain, we also craft various empirical metrics that measure the *importance* of frequency components under various paradigms.

### 5.1 IMPORTANCE BY VULNERABILITY

We measure the importance of a frequency component by measuring the attack success rate when an adversarial attack is constrained to frequency $f$. Essentially, we are quantifying the importance by measuring the expected vulnerability of each frequency. This amounts to measuring the accuracy of $h(x + \delta_f)$, where $\delta_f$ is the adversarial perturbation that is constrained to frequency $f$, obtained using the aforementioned DCT-PGD method. A lower accuracy of the model for a particular $\delta_f$ indicates a more *important* frequency $f$. In Figure 2, we visualize the accuracy of models with both normal training and adversarial training across different datasets under this setting. We see that only in the case of CIFAR-10, the trends for normal training and adversarial training are reversed, indicating that attacks constrained to higher frequencies are more successful for normal models, while lower frequency attacks are more effective on the adversarially trained models. In TinyImageNet and ImageNet datasets, we see that the overall trend remains same across the two training paradigms with adversarial training improving robustness across the spectrum. To obtain a high level view, we design another set of experiments where instead of attacking individual frequency components, we restrict the attack to frequency ranges (or bands, set of 16 equal divisions of the spectrum). The results of these under DCT-PGD version of Auto-Attack are shown in Figure A.7. In their work, Wang et al. (2020b) had claimed that low frequency perturbations cause visible changes in the image, thus defeating the purpose of *imperceptibility* clause of adversarial examples. However, we find that for

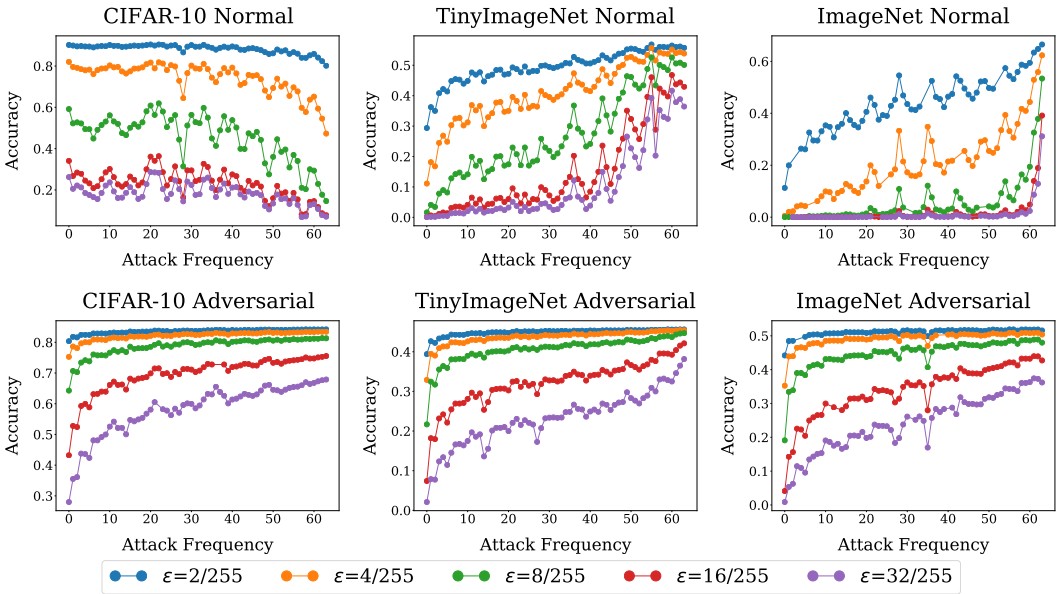

Figure 2: Vulnerability scores (Accuracy under attack) visualized per frequency across datasets. Notice that the trends are reversed from normal training to adversarial training in the case of CIFAR-10. The results for different frequency bands, under Auto-Attack is shown in Figure A.7.

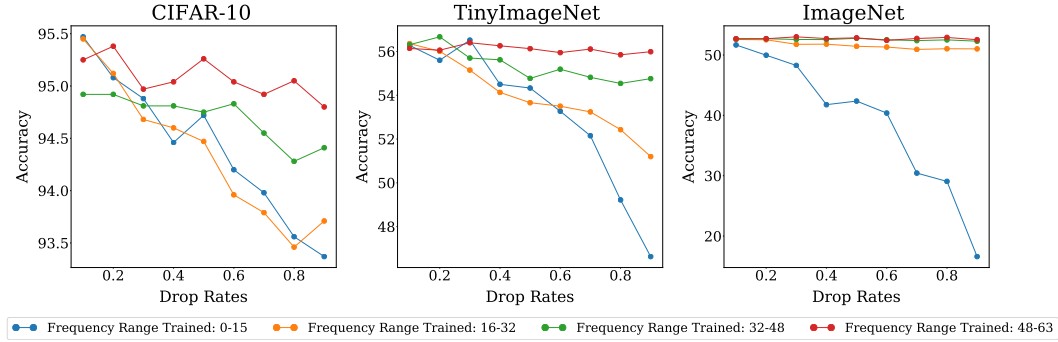

Figure 3: Accuracy for models trained with varying drop rates, for different frequency ranges.

larger datasets, such perturbations are imperceptible to a human. Example images have been shown in Appendix (Figure A.20 and A.21).

## 5.2 IMPORTANCE DURING TRAINING

With the objective of understanding the relative importance of frequency components while training, we formulate an experiment where we train models by masking out (making them zeros) frequency components of the input in a probabilistic manner and then using the trained model for normal inference. Example images when certain frequency bands are dropped is shown in Figure A.18. We train four types of models, where the frequency masking is restricted to four equal frequency bands and the amount of masking/dropping is controlled by a parameter $p$. This translates to training

$$\arg\min_{\theta} \mathcal{L}(h(x_{\hat{f}}; \theta), y) \tag{17}$$

$$\text{where } x_{\hat{f}} = D^{-1}(M \odot D(x)) \tag{18}$$

$$\text{and } M_z = \begin{cases} 1 & z \sim \mathcal{U}_p \wedge z \in [f_1, f_2..., f_k] \\ 0 & \text{else} \end{cases} \text{ is the Mask generated using } p \tag{19}$$

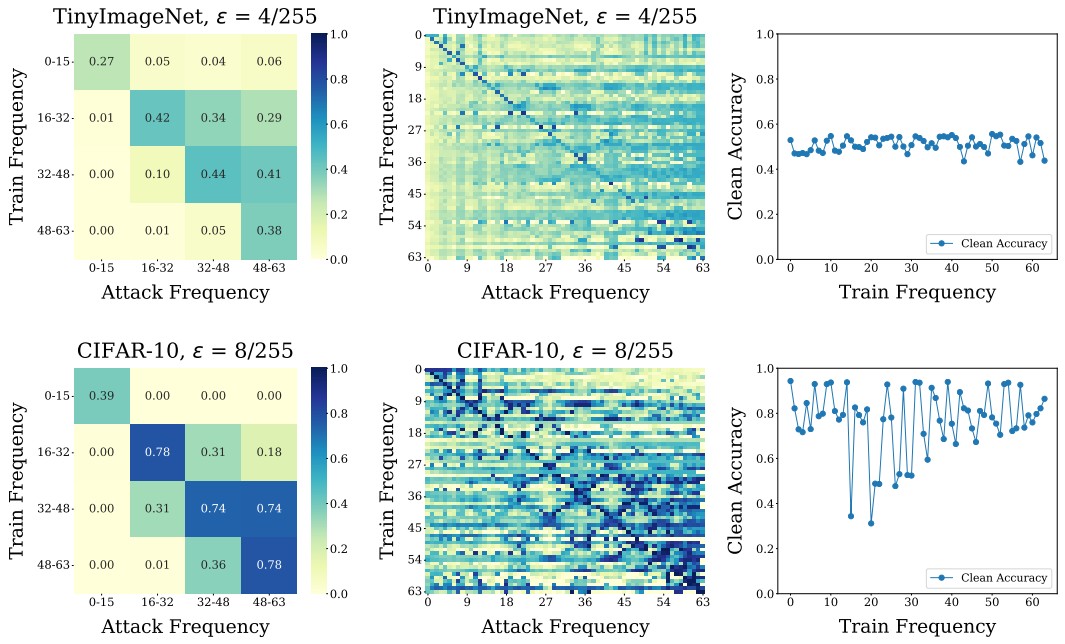

Figure 4: Frequency-based adversarial training across datasets. In the first column we show the results of adversarially training and testing for different frequency ranges. Next, we show results of the same experiments across individual frequencies. The last column shows clean accuracy for each frequency.

where $x_{\hat{f}}$ is the input constrained to a particular frequency band within the range $[f_1, f_2, \cdots f_k]$. While training, we select the frequencies to be dropped using a random uniform distribution $\mathcal{U}$, with the percentage of dropping controlled by parameter $p$. A value of $p = 1$ indicates all frequencies in the specified band are set to zero. We train a total of 36 models per dataset, encompassing 9 different drop rates ($p$ values) and 4 frequency bands. The experiment is repeated across datasets and the results are shown in Figure 3. As expected, we observe that a higher drop rate leads to lower accuracy. We also see that across datasets, high drop rates in low frequency band of 0-15 affects the model more. This behaviour is expected as lower frequencies have a strong relation with the labels (Wang et al., 2020b) and their extreme dropping leaves the model with little information to learn from. But if we observe the degree to which it affects the performance, we see disparities between the datasets. For example, the model trained on CIFAR-10 experiences a mere ∼2% drop even when 90% of frequencies in the low band (frequencies 0-15) are dropped. Under the same condition, the model on TinyImageNet experiences ∼10% drop and the model on ImageNet experiences a whopping ∼35% drop in accuracy, highlighting the relative importance of these frequency bands. Also, note how very high drop rates in the highest frequency bands (frequencies 48-63) have little to no effect in non CIFAR-10 models.

## 6 ADVERSARIAL TRAINING WITH FREQUENCY-BASED PERTURBATIONS

Till now, we have analyzed the frequency properties of the model across datasets. In all experiments so far, we merely observed how the model *reacts* to adversarial perturbations under various frequency constraints. To further understand the properties of robustness in the frequency domain, we propose to train models with adversarial perturbations restricted to these frequency subspaces, a first of its kind. The training follows

$$\min_{\theta} \max_{||\delta_f||_p \leq \epsilon} \mathcal{L}(h(x + \delta_f; \theta), y) \tag{20}$$

where $\delta_f$ is adversarial noise restricted to a frequency subspace defined by $f$. To obtain a high-level view of the process, we first train models adversarially with frequencies restricted to four equal

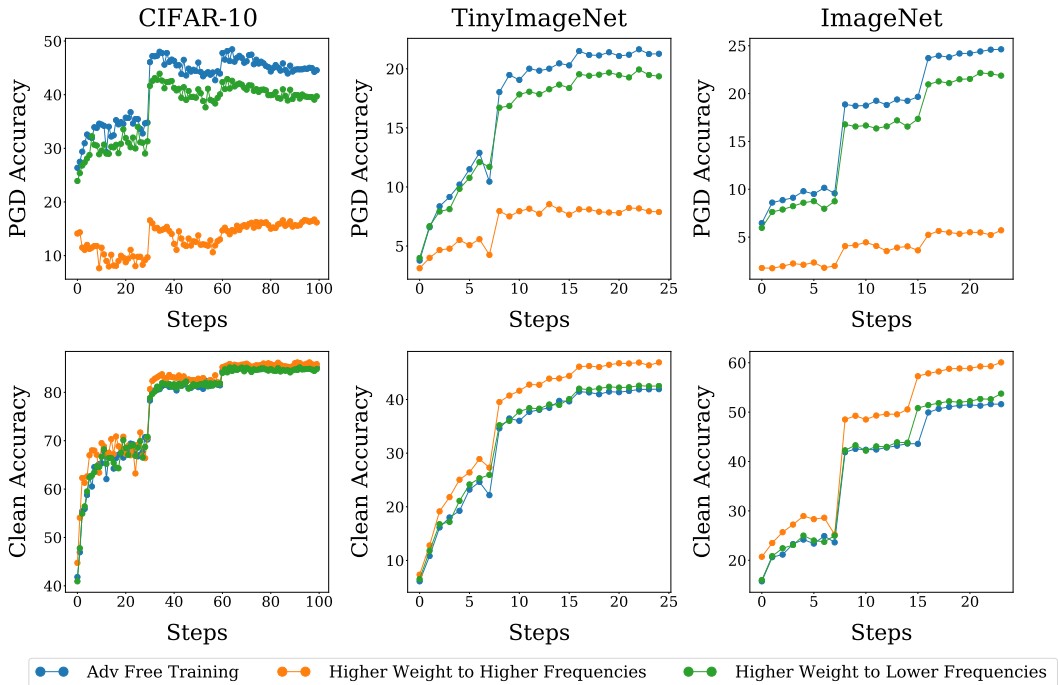

Figure 5: Illustration of unequal epsilon distribution. Here we see that models where low frequency perturbations are favoured ends up with higher robustness, but lower clean accuracy.

frequency bands, ranging from low to high. Predictably, the models perform well when adversarial PGD attack is also restricted to the same frequency bands. The resulting robustness heatmap of attacks across the spectrum is shown in first column of Figure 4. For a more fine-grained view of the same, we adversarially train 64 models for each dataset, by perturbing each individual frequency. Then we adversarially attack these models in every frequency to produce a robustness heatmap, shown in the second column of Figure 4. In their work, Yin et al. (2019) had claimed that training with low-frequency perturbations did not help the model to be robust against those frequencies. Their analysis was not based on adversarial perturbations, but their claim was generalized. This effect was not observed in our experiments. We see that the model has good robustness when trained and tested against low-frequency perturbations, across datasets. The diagonals of the robustness heatmaps tell us that models perform well against an adversary constrained to the same frequency used for training. Moreover, we also see that models trained with perturbations restricted to mid/higher frequencies can withstand attacks from a fairly broad range of frequencies compared to models trained with lower frequency perturbations. Now that we have established this new training paradigm, we explore its various nuances and intriguing properties.

## 6.1 THE UNEQUAL EPSILON DISTRIBUTION

*Do all frequencies have the same impact in adversarial training?* To answer this question, we modify the construction of adversarial perturbation $\delta$ by weighing contributions from different frequency components and manipulating the value of $\epsilon$ they receive. It follows

$$\delta = \sum_{i=0}^{K} \eta_i \cdot \text{sgn}(\nabla_x \mathcal{L})_i \text{ for } L_\infty \text{ norm} \tag{21}$$

$$\eta_i = \frac{\epsilon}{K - i} \tag{22}$$

where $K$ is the number of equal frequency bands (four in our case) and $\eta$ is a linear scaling parameter. This setting effectively translates to giving more importance to perturbation in one frequency space over the other. We train 2 models: One as described by equation 22, favoring lower frequency bands and then its complement, by reversing $\eta$ and favoring higher frequency bands. For these ex-

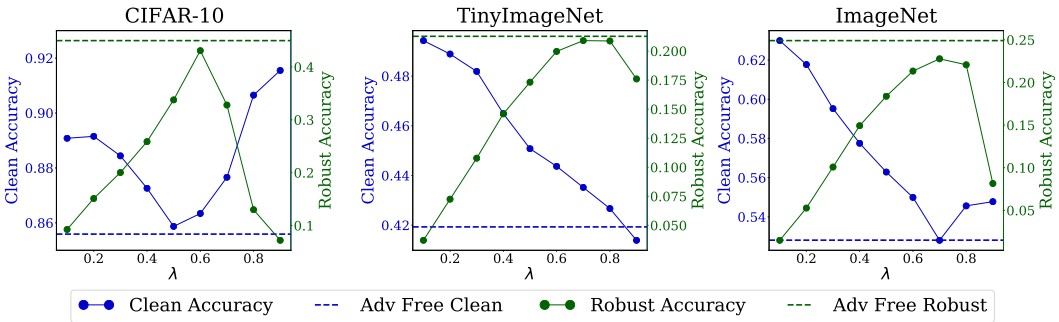

Figure 6: Clean Accuracy vs Robustness across datasets, compared with standard adversarial training for free method. Note that the Y-axis scales are different. Here $\lambda$ controls the weight of adversarial perturbation towards lower frequencies.

periments, we employ Free adversarial training by Shafahi et al. (2019). The plot of PGD and clean accuracy during training are shown in Figure 5. We see that the model in which lower frequencies are favoured acts closest to standard PGD-based adversarial training. This shows that for a model to be robust, it only needs to be adversarially trained in the frequencies that matter most and not the entire spectrum. But at the same time, we see that the model where high frequency perturbations are favoured shows superior clean accuracy in all datasets except CIFAR-10. These results tell us that frequency based perturbations are intricately tied with clean accuracy and robustness of a model. We explore this in detail in the next section.

## 6.2 ACCURACY VS ROBUSTNESS: AN ALTERNATIVE PERSPECTIVE

Building on top of previous results, we design an experiment to examine the accuracy vs robustness trade-off that is commonplace while training robust models. We introduce a parameter $\lambda$ that controls the weight given to frequency components in the perturbation during adversarial training. The update step for PGD under $L_\infty$-norm now looks like:

$$\delta = \lambda \cdot \left[ \alpha \cdot \text{sgn}(\nabla_x \mathcal{L}_{\text{LF}}) \right] + (1 - \lambda) \cdot \left[ \alpha \cdot \text{sgn}(\nabla_x \mathcal{L}_{\text{HF}}) \right] \tag{23}$$

where $\nabla_x \mathcal{L}_{\text{LF}}$ and $\nabla_x \mathcal{L}_{\text{HF}}$ are gradients restricted to low (frequencies 0-31) and high frequencies (frequencies 32-63) respectively. We adversarially train ten different models by varying the value of $\lambda$ and show their clean and robust accuracy in Figure 6. We see that in the case of TinyImageNet and ImageNet, the clean accuracy decreases when we train with low frequency perturbations, while increasing robustness. In case of CIFAR-10, we see that there is an initial increase in robustness followed by a steep fall. This is because higher frequencies have a significant role in adversarial robustness for this dataset, which is not achieved when $\lambda$ values are high. We also observe a steep fall in robustness for ImageNet at $\lambda$ of 0.9. This is because the frequency importance is distributed in the low-mid range for ImageNet (Figure 1a) and very high $\lambda$ values tend to ignore the 32-48 frequency bands. These results establish that robustness and clean accuracy of an adversarially trained model are dependent on the frequencies we perturb. The $\lambda$ parameter gives us control over the trade-off, enabling us to be more prudent while designing architectures and training regimes that demand a mix of clean accuracy and robustness.

## 7 CONCLUSION

In this paper, we analyze adversarial robustness through the perspective of spatial frequencies and show that adversarial examples are not just a high frequency phenomenon. Using both theoretical and empirical results we show that constituent frequencies of adversarial examples are dependent on the dataset. Then we propose and study the properties of adversarial training using specific frequencies, which can be used to understand the accuracy-robustness trade-off. These results can be utilized to train robust models more quickly by focusing on the frequencies that matter most. We hope that our findings will resolve some misconceptions about the frequency content of adversarial examples and aid in creating more robust architectures.

## 8 ETHICS

Adversarial examples pose a unique challenge to real world deep learning systems. We believe that our analysis will aid in the development of adversarial attacks as well as robust architectures. While we are aware of the potential for malicious uses of both of these applications we find minimal direct ethical concerns with the work in this paper. It is our hope that our work will only provide a deeper understanding of what constitutes an adversarial example and the mechanisms behind adversarial training in order to guide future research in this area.

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

# A APPENDIX

## A.1 PROOFS

Here are the proofs for some results from above. In equation 11 we mentioned

$$\nabla_\delta Y = \nabla_x Y = \nabla_{\hat{x}} Y \tag{24}$$

Consider a neural network $y = h(x; \theta)$. Let the adversarial sample be $\hat{x} = x + \delta$, where $\delta$ is the additive adversarial noise.

$$y = h(\hat{x}) = h(x + \delta) \tag{25}$$

$$\frac{dy}{dx} = h(x + \delta)' \cdot 1 = \frac{dy}{d\delta} \tag{26}$$

$$\frac{dy}{d\hat{x}} = h(\hat{x})' = h(x + \delta)' \text{ hence} \tag{27}$$

$$\frac{dy}{d\hat{x}} = \frac{dy}{d\delta} = \frac{dy}{dx} \text{ or } \nabla_\delta Y = \nabla_x Y = \nabla_{\hat{x}} Y \tag{28}$$

In the same section's equation 12 we also mentioned $\nabla_x L \propto \nabla_x Y$.

$$\text{Let } L = \frac{1}{2} \left( h(x; \theta) - \hat{y} \right)^2 \text{ be the loss.} \tag{29}$$

$$\frac{dL}{dx} = (h(x; \theta) - \hat{y}) \cdot h(x; \theta)' \tag{30}$$

$$\text{here } h(x; \theta)' = \frac{dy}{dx} \text{ and } (h(x; \theta) - \hat{y}) \text{ is a constant} \tag{31}$$

$$\frac{dL}{dx} = K \cdot \frac{dy}{dx} \text{ which implies} \tag{32}$$

$$\nabla_x L \propto \nabla_x Y \tag{33}$$

## A.2 TRAINING DETAILS

We utilize ResNet-18 in all our experiments (unless stated otherwise). For ImageNet and TinyImageNet datasets, we train for a total of 100 epochs, with an initial learning rate of 0.1 decayed every 30 epochs, momentum of 0.9 and a weight decay of 5e-4. In Madry adversarial training for the same, we use an $\epsilon$ value of 4/255. Under adversarial training for free setting, we train both models for 25 epochs with learning rate decayed every 8 epochs and the $m$ (repeat step) set to 4.

For CIFAR-10, we train the model for total of 350 epochs, starting with a learning rate of 0.1, decayed at 150 and 250 epochs and use the same setting with an $\epsilon$ of 8/255 for Madry training. In adversarial training for free setting, we train the model for 100 epochs with learning rate decay every 30 epochs and the $m$ value set to 8.

We utilize the pretrained models provided by PyTorch for ImageNet normal models. All experiments involving ImageNet-based adversarial training were done using Adversarial training for free method, with total epochs of 25 and $m$ value set to 4.

## A.3 FREQUENCY RANGE-BASED PERTURBATIONS

We revisit the results shown in Figure 2 and show the same in a broader sense by attacking different frequency ranges. The results under DCT-PGD based Auto-Attack are shown in Figure A.7. We can see that the trends which were observed and discussed in earlier sections remain unchanged.

## A.4 WHAT DO FREQUENCY ATTACKS TARGET ?

A natural question that might arise with respect to DCT-PGD paradigm is how can we be sure that there is proportionate distortion in the frequency space as well. ((Rephrase)) We can visualize this using simple properties of the DCT. Consider the 1-D DCT from above. Since it is a linear transform, we can rewrite it as :

$$D(z) = WZ \text{ where W is the linear DCT transform on the input tensor Z} \tag{34}$$

$$\hat{x} = x + \delta \text{ in DCT space becomes} \tag{35}$$

$$W \cdot \hat{x} = W \cdot x + W \cdot \delta \tag{36}$$

$$\tag{37}$$

The elements of $W$ represent different standard DCT basis functions, such that the lower frequencies are in upper left corner and the higher frequencies are towards the lower right corner. For any

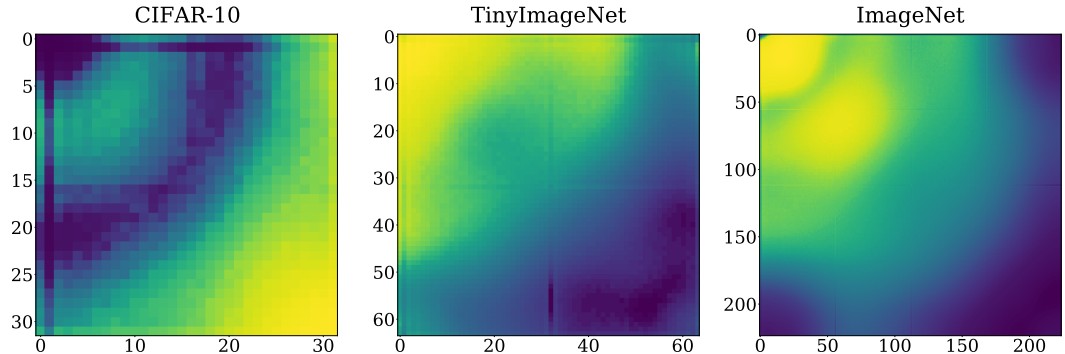

Figure A.8: Noise gradients visualized under L2 attack for normally trained ResNet-18 models. We used attack $\epsilon = 1$ for all models.

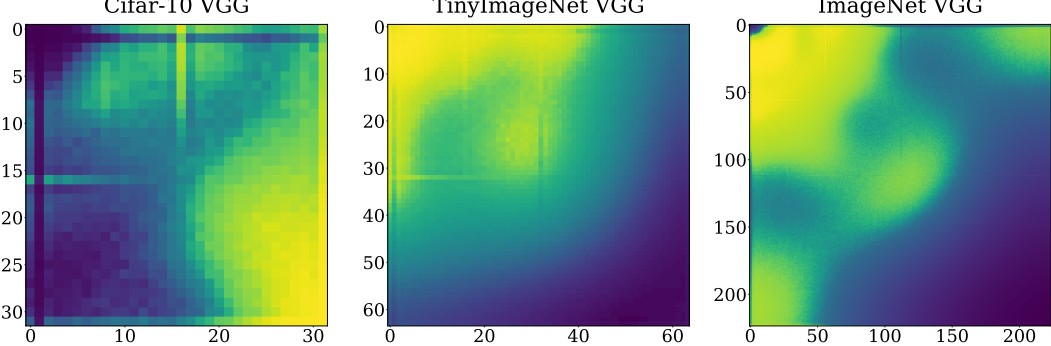

Figure A.9: Average Noise Gradients of VGG-16 models, across datasets

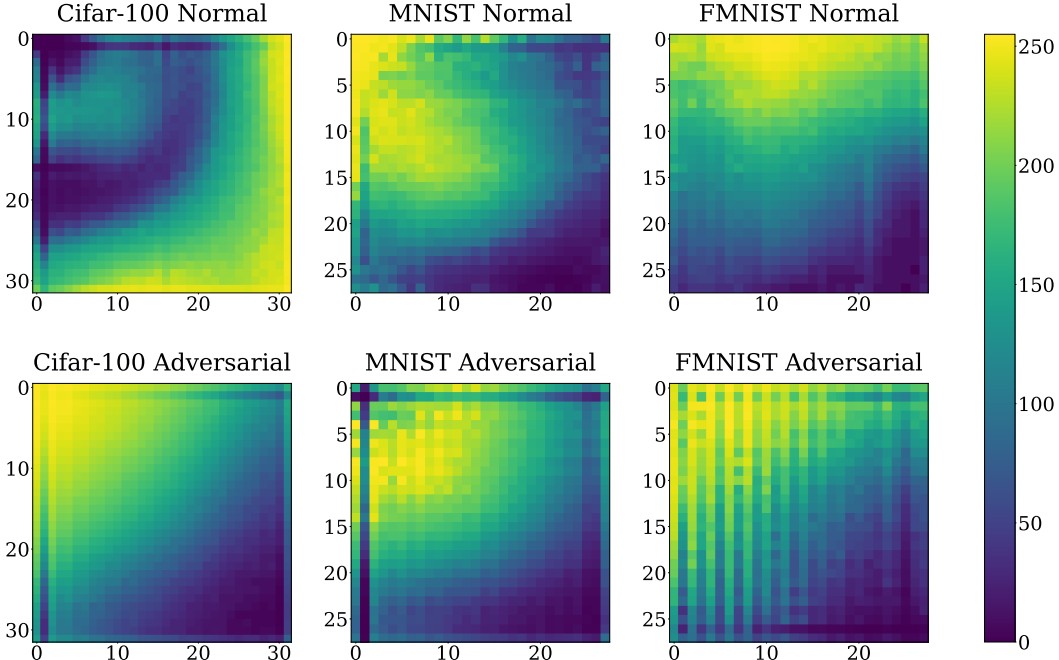

Figure A.10: DCT of Average Noise Gradients across additional datasets

element $i$ that also represents a frequency component, we can say that:

$$W_i \cdot \hat{x}_i = W_i \cdot x_i + W_i \cdot \delta_i \tag{38}$$

Essentially, we see that in the frequency space, each component of the resulting adversarial example $\hat{x}$ is linearly distorted by the corresponding frequency component of noise $\delta$.

## A.5 DOES MODEL MATTER?

We run the experiments across VGG-16 to confirm that the above trends are model agnostic and aren't just limited to ResNet style architectures. In the results shown in Figure A.9 we see that the trends remain unchanged across datasets.

## A.6 DOES IMAGE SIZE MATTER?

To confirm that the anomalies of adversarial examples are indeed due the underlying dataset and not just the size, we repeat the experiment by training models where ImageNet and TinyImagenet images are resized to smaller sizes using bicubic filter. The average noise gradients calculated from these models are shown in Figure A.15.

## A.7 L2-BASED ADVERSARIAL ATTACKS

We repeat the same experiments to calculate noise gradients under the L2 attack. We do not observe any divergent behaviour, compared to $L_\infty$ attack. The results across datasets are shown in Figure A.8.

## A.8 EXTENDING TO MORE DATASETS

We also repeat the experiments across datasets, including non-ImageNet derived datasets like MNIST, Fashion-MNIST and CIFAR-100.The results are shown in Figure A.10.

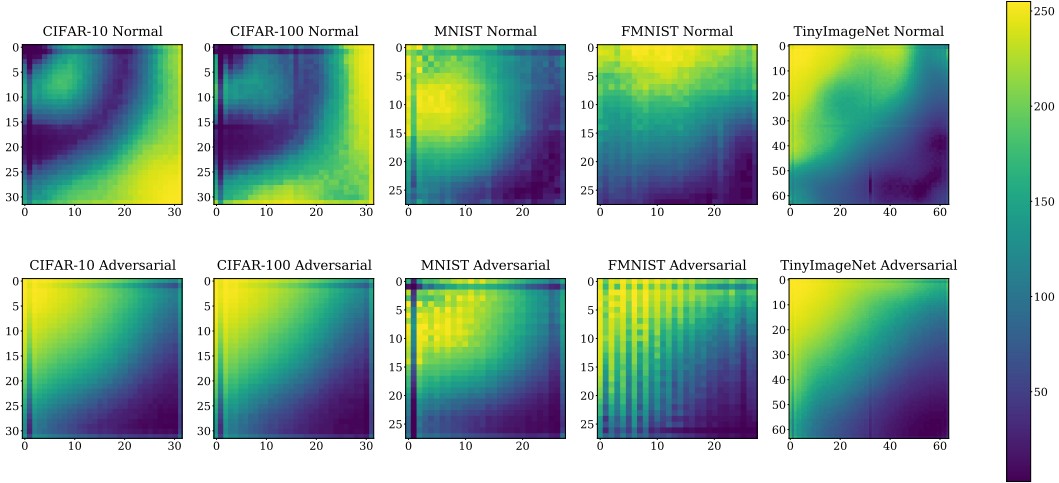

Figure A.11: DCT of Average Noise Gradients with Auto-Attack

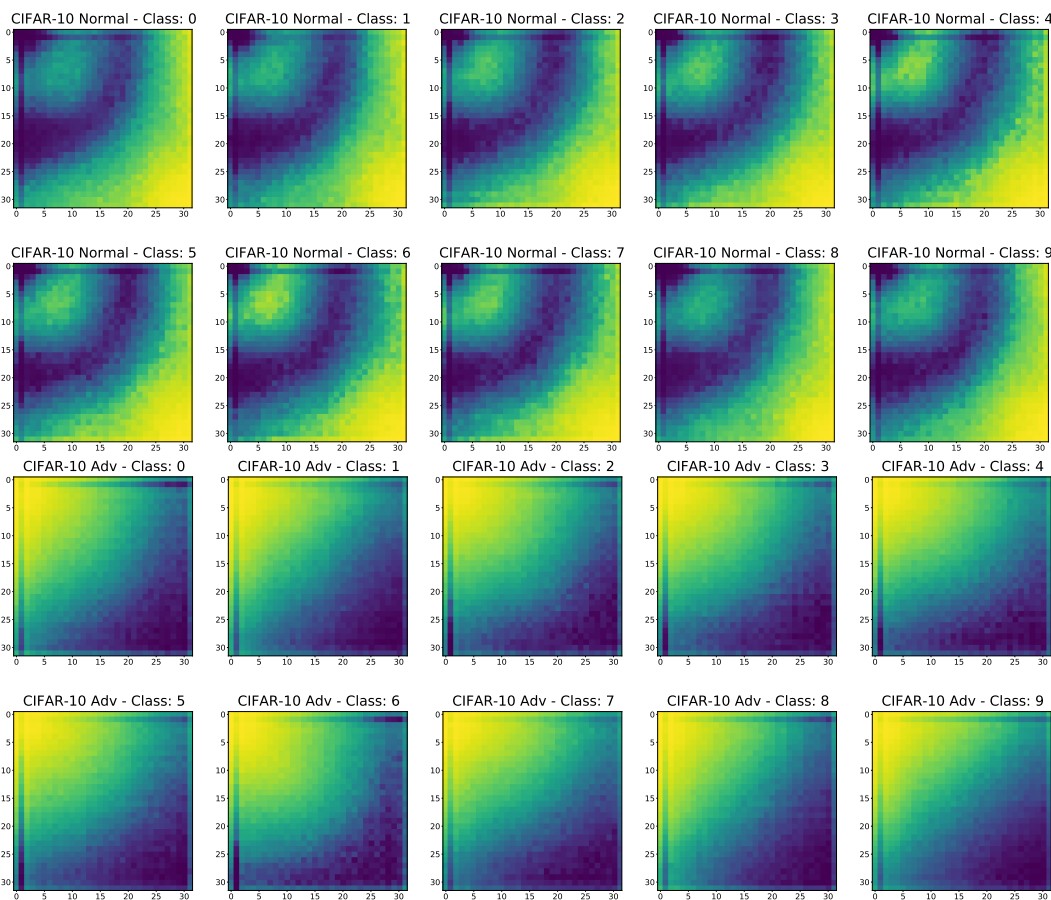

Figure A.12: DCT of Average Noise Gradients Classwise for CIFAR-10

## A.9 EFFECT OF AUTO-ATTACK

We calculate and plot the noise gradients for all models under Auto-attack setting. In general, there appears to be no significant difference when compared to results from PGD attack. The results are shown in figure A.11.

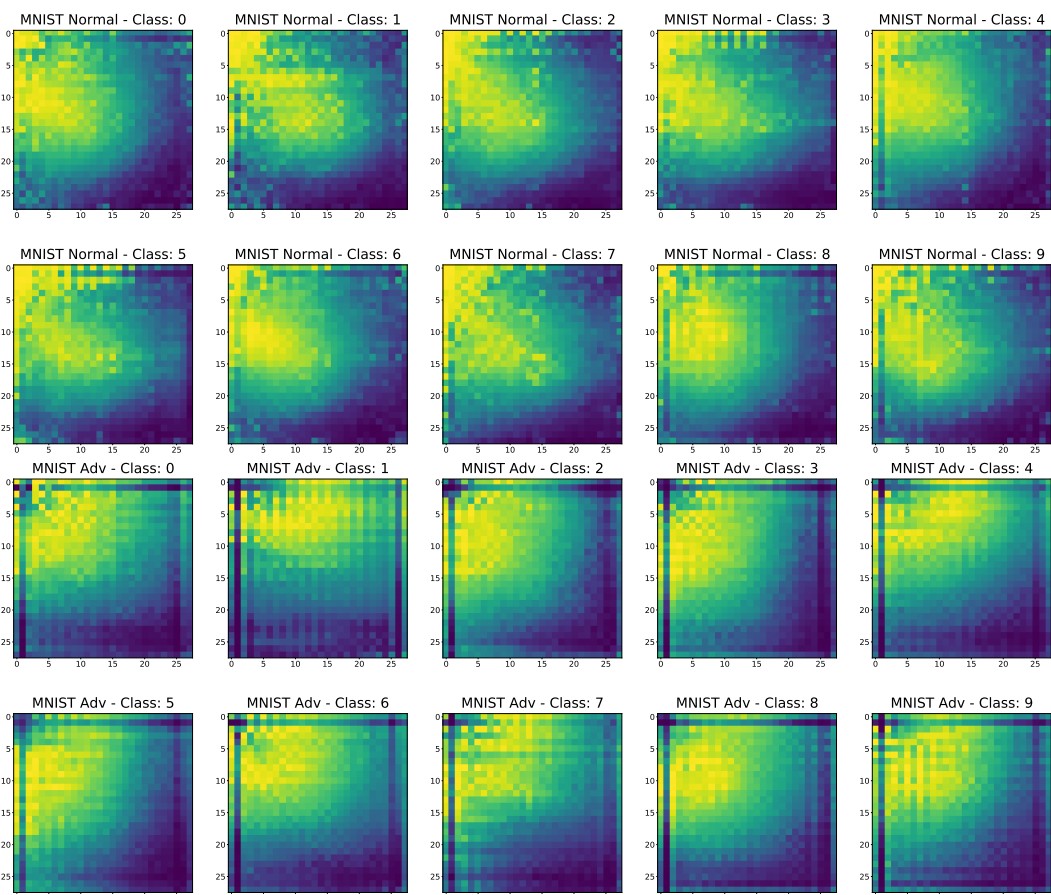

Figure A.13: DCT of Average Noise Gradients Classwise for MNIST

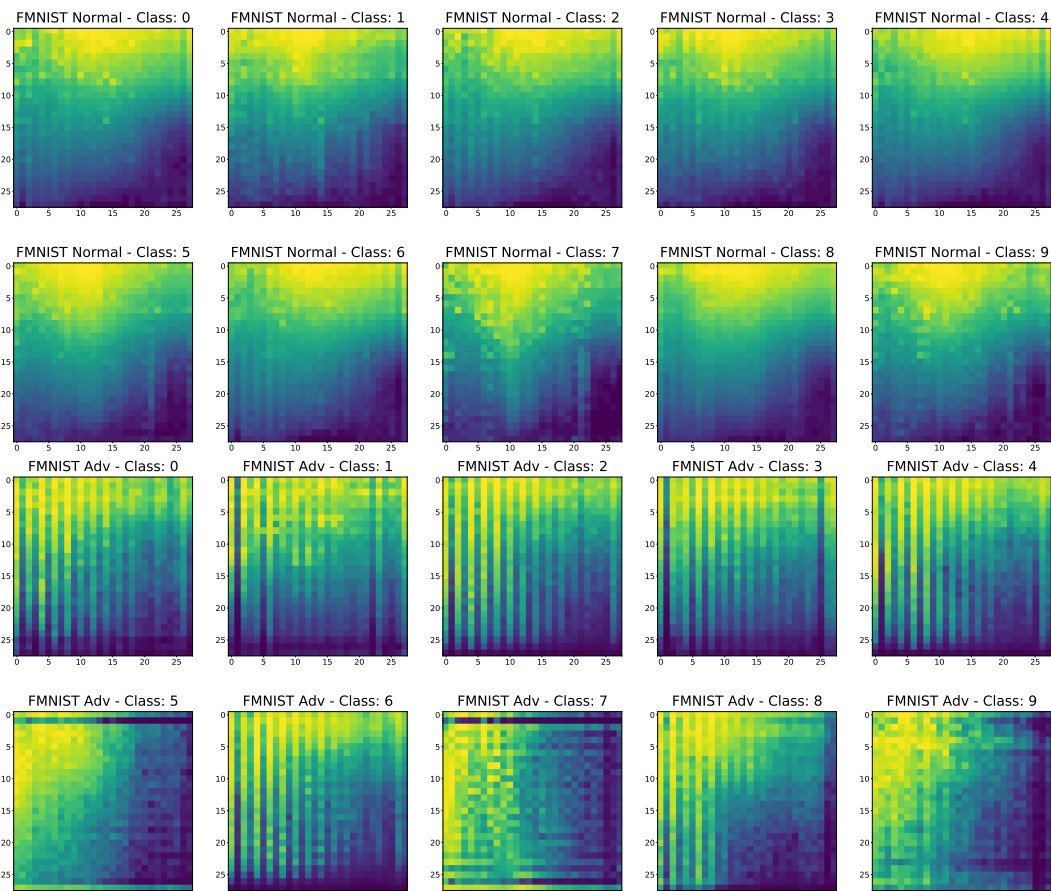

Figure A.14: DCT of Average Noise Gradients Classwise for Fashion-MNIST

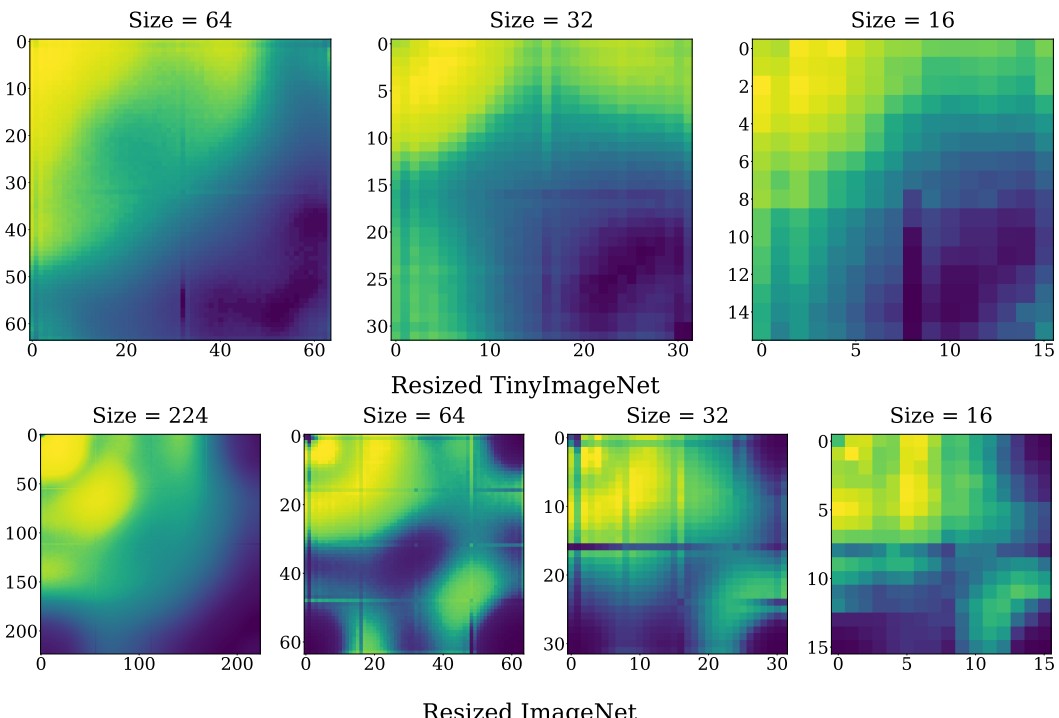

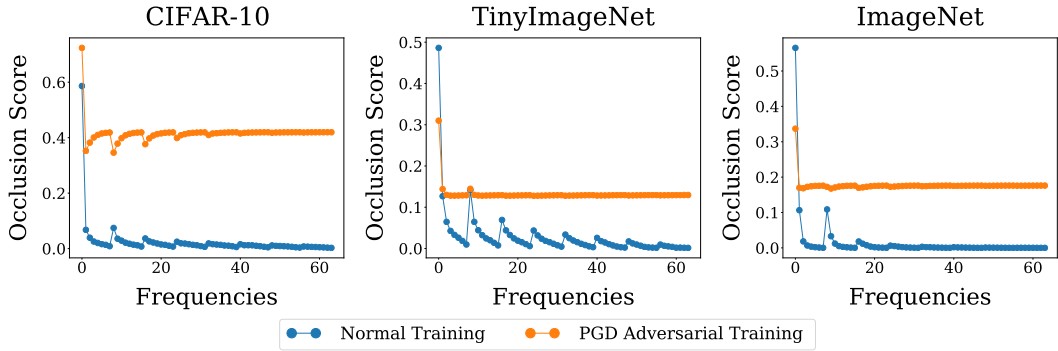

Figure A.15: Effect of resizing the images on TinyImageNet and ImageNet.

Figure A.16: Occlusion Scores averaged over validation, across three datasets. Adversarial training is with $L_\infty$ norm with $\epsilon$ of 8/255 for CIFAR-10 and 4/255 for others.

## A.10 CLASS WISE RESULTS

We also investigate if there exists different frequency distribution for each class in a dataset. We show these results for CIFAR-10 (Fig A.12), MNIST (Fig A.13) and Fashion-MNIST (Fig A.14) datasets, for both normally trained and adversarially trained models. Apart from subtle differences, we do not see any general shift in the trends and observations.

## A.11 OCCLUSION SCORE

We borrow a simple metric the "Occlusion Score" from Wang et al. (2020c). Given a network $h(x)$ and an image $x$, the occlusion score $O_f(x)$ for frequency $f$ on class $c$ is defined as:

$$O_f(x) = |h(x)^c - h(x_{\hat{f}})^c| \tag{39}$$

where $x_{\hat{f}}$ refers to the input image $x$ with the frequency $\hat{f}$ removed from the spectrum. A higher score indicates that there is a drop in model accuracy when that particular frequency is removed,

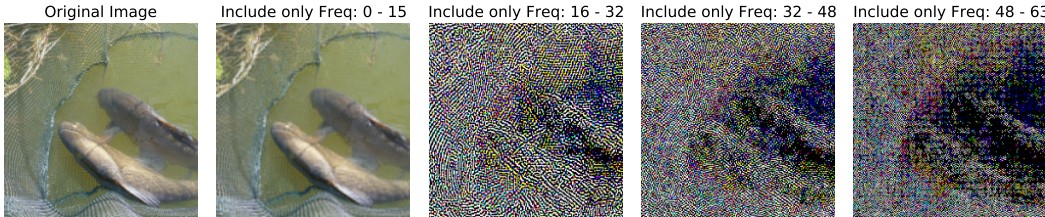

Figure A.17: ImageNet examples where image is reconstructed using only specified frequency bands

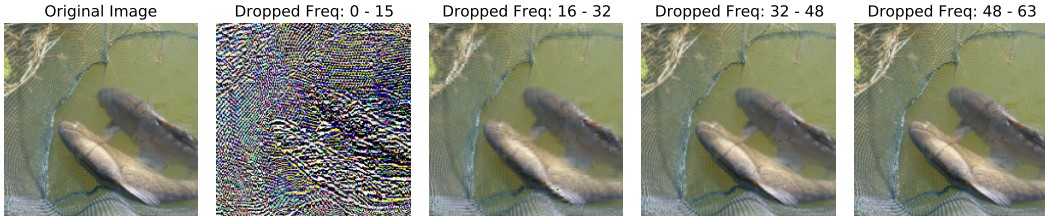

Figure A.18: ImageNet examples where image is reconstructed after dropping (zeroing) certain frequency bands

which implies the importance of the frequency. In their paper, Wang et al. (2020c) show the results of this metric on CIFAR-10 dataset and incorrectly conclude that adversarial training tends to shift attribution scores from higher frequency regions to lower frequency regions. We show that this is not the case by simply extending the experiment to include TinyImageNet and ImageNet datasets From the results shown in A.16, we can clearly see that in non CIFAR datasets, the attribution scores are already skewed towards lower frequencies and the shift after adversarial training happens across all frequencies.

## A.12 EXAMPLES OF FREQUENCY-BASED PERTURBATIONS

We show example images under different perturbation budgets of $L_\infty$ norm, across datasets in Figures A.19, A.20 and A.21. We also show examples of images when certain frequency bands are dropped A.18 and the complementary case of including only specified frequencies A.17.

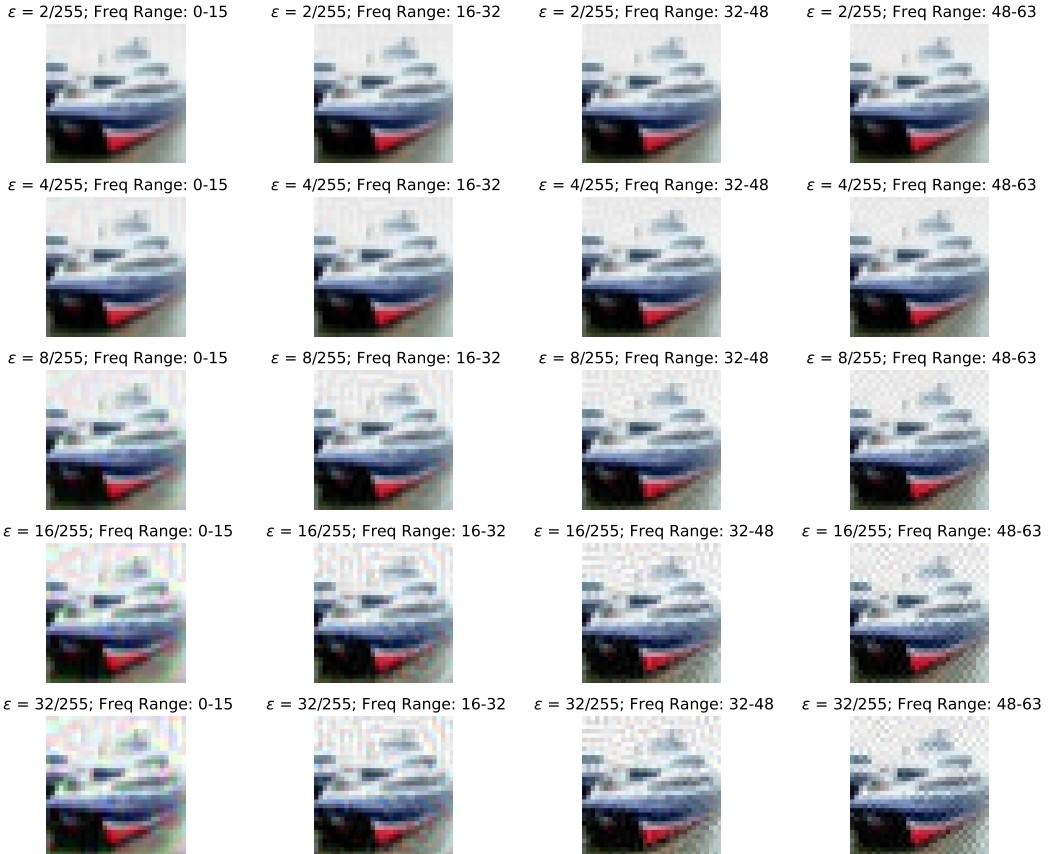

Figure A.19: CIFAR-10 example images under different attack settings.

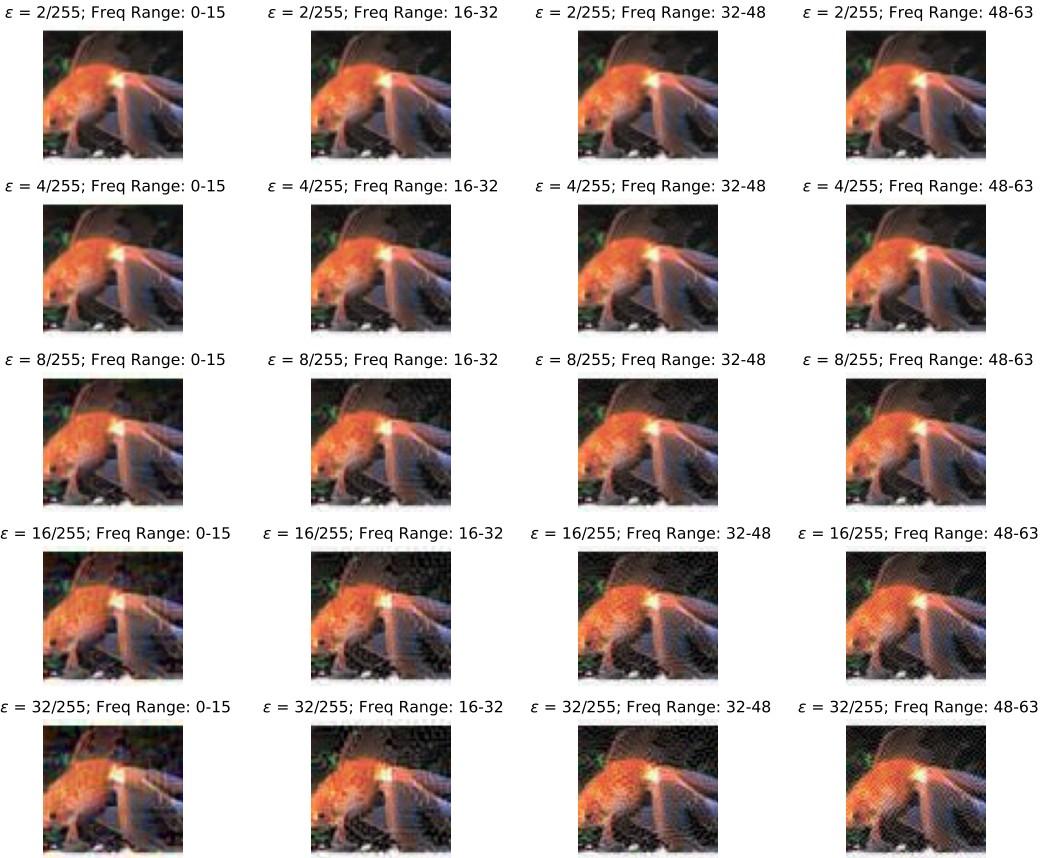

Figure A.20: TinyImageNet example images under different attack settings.

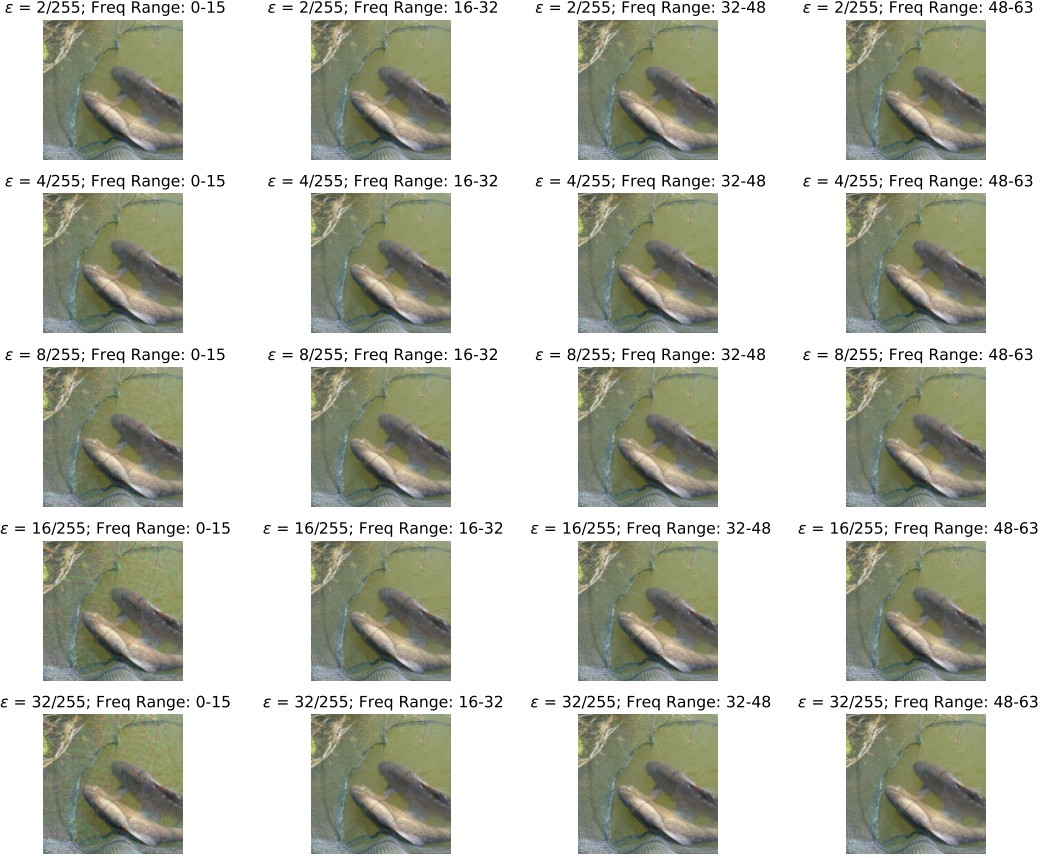

Figure A.21: ImageNet example images under different attack settings.

