# OpenReview forum: "A Frequency Perspective of Adversarial Robustness"
_ICLR.cc/2022/Conference — ICLR 2022 Submitted_

### Official Review · Reviewer_dU9j · 2021-10-22

**Correctness:** 4
**Technical Novelty And Significance:** 3
**Empirical Novelty And Significance:** 3
**Recommendation:** 8
**Confidence:** 4

**Main Review:**

# Strong points
* This work provides valuable insights to the adversarial machine learning community regarding the frequency-properties of adversarial examples and I believe other researchers in the field will find the insights in this work useful. In fact, I previously observed similar phenomena in my personal research.
* The paper is well written, easy to follow, and provides extensive supplementary material.
* The experimental design in section 6.1 (Weighing the contributions of different frequency components) is a nice idea providing interesting results.
* Similarly, the results from section 6.2 are interesting and give new insights for the adversarial training community.
* The authors performed extensive experiments to support their claims, even providing many results on ImageNet.

# Weak points
* In some sense, the finding that the frequency properties of adversarial perturbations are dependant on the dataset is somewhat common sense. This work mainly explores adversarial examples for models trained on natural images, which are generally more low frequency. I assume that the frequency property of adversarial examples would be different for models trained on relatively more high-frequency images, such as radar signals or vibration signals. Such an experiment could also be insightful to further support one of the main points of this work.
Additionally, it would have been interesting to see in addition to Figure 1, in which frequency range the natural images are located. This could allow a comparison of the frequency properties of adversarial examples to the natural images and maybe insights if adversarial perturbations exploit different frequency areas or overlapping frequency areas of the natural images.
* The authors solely examine the frequency properties of adversarial perturbations for ResNet18 (and VGG16 in Figure A.9), however, other influence factors are not considered, such as other network architectures (DenseNet, ViT, MLP-Mixer), optimization properties (learning rate, optimizer choice). Hence, this work also provides only a partial view of the frequency perspective of adversarial robustness. The authors should discuss or point out this limitation of their work.
* [1] also observes that “In low-res datasets, higher frequencies tend to be disproportionately important, but the effect is less prominent on high-res ones.”, which is similar to the finding in this work: “We see that for normally trained CIFAR-10 models, the DCT of noise gradient activations are towards the higher frequencies [...] Whereas for TinyImageNet and ImageNet models, we observe that the activations are already in lower-mid frequencies”. This slightly limits the provided insight of this work. The authors should discuss the differences.

[1] Dissecting the High-Frequency Bias in Convolutional Neural Networks; CVPRW 2021


**Summary Of The Paper:**

This work explores adversraial robustness from a frequency perspective. While this had been done before, this work challenges the commonly held notions, that adversarial perturbations are mainly a high-frequency phenomenon. The authors provide insight that the frequency properties of adversarial examples are dependant on the underlying training dataset. Additionally, the impact of different frequency properties in adversarial training is explored.

**Summary Of The Review:**

Accept. This work provides beneficial insights to the community and the strong points outweigh the weaknesses of this work.

---

> ### Author Response · Authors · 2021-11-13
> **Response to Reviewer dU9j**
>
> We thank R4 for finding our insights beneficial for the community and for valuable feedback.
> * R4 is indeed correct in pointing out that most natural images tend to be in the low frequency regime and further experiments on radar/vibration images can definitely enhance our understanding of adversarial examples. However, this study focuses on natural images in standard datasets. We will add a discussion on other types of images.
> * We thank R4 for suggestion on presenting results using other model architectures. We have now attached results on DenseNet-121 and ViT models (for ImageNet) in the supplementary material. We do not find any significant differences in results on vision transformer-based models. Further studies on optimization properties will definitely broaden our current understanding.
> * We thank R4 for pointing out the work done by Antonio et al. We have acknowledged and discussed their work in the updated version of the paper.

---

### Official Review · Reviewer_pNyu · 2021-11-02

**Correctness:** 2
**Technical Novelty And Significance:** 2
**Empirical Novelty And Significance:** 2
**Recommendation:** 3
**Confidence:** 3

**Main Review:**

Generally speaking, I think the author is very knowledgable for the corresponding field of adversarial robustness. They put a lot efforts to analyze the intriguing phenomena of adversarial examples. And I also agree with using the perspective of frequency  to understand this unsolved problem.

However, the content of this paper really confuses me. Currently, I cannot tell the level of its contribution. In particular, the author list three contributions in the 1st Section. Yet, most content in this section is about related works. The author only bring up their own work at the end of this section with three very brief descriptions. The first contribution (neither high nor low) is somewhat interesting but not well-supported. The author explain this argument in Section3 but I cannot understand their claim. What do you mean by "as one cannot verify the con- verse setting of blocking low frequency components. This is because low frequency components are inherently tied with labels (Wang et al., 2020b), conflating the two phenomena." Is there any other evidences that can support your claim? For the other two contributions, I think the authors themselves did not know how their work can help the community. Simply introducing the Fourier perspective into adversarial robustness or defenses is not enough to become a good work. After all, adopting the Fourier perspective is not the original idea of this paper. To help others understand the actual contribution, the authors should provide more details about how the analyses in this paper can help existing research and future works. Currently, I cannot find such claims. If there is any, the author should highlight these parts during rebuttal or maybe put them into the 1st Section.

**Summary Of The Paper:**

This paper intends to investigate and analyze the phenomena of adversarial examples through the perspective of Fourier Analyses. It claims three major contributions:
1) they claim that adversarial examples is neither high frequency nor low frequency.
2) they adapt adversarial training with Fourier Analyses
3) they provide a new framework to measure robustness

**Summary Of The Review:**

I decide to reject this paper for now. However, if the author can properly highlight their contributions during rebuttal, I may well reconsider my recommendation.

---

> ### Author Response · Authors · 2021-11-13
> **Response to Reviewer pNyu**
>
> We thank R3 for the feedback.
> * We actually utilize the Discrete Cosine Transform (DCT) basis and __not__ "`Fourier" transforms for our analysis. They are mathematically quite different. We hope this clarification helps alleviate some of R3's concerns.
> * We believe that an extensive related work section is extremely relevant to our work as this is a niche topic, which has been prone to misinterpretations and confusion in the past.
> * For our claim about adversarial examples being "neither high frequency or low frequency", we not only show it via perturbation gradient analysis, but also support it with rigorous empirical results, spanning different datasets presented in Sections 5.1 and 5.2. We follow up these results by repeating the experiments across different architectures (Appendix Section A5), different attack settings (Appendix Section A7, A9) and image size ablations (Appendix Section A6). We believe all these results offer solid and strong support to our claims.
> * Regarding R3's doubts about "we cannot verify the converse setting of blocking low frequency components" -- we wanted to draw attention to the fact that while "blocking high frequency" components was considered a standard method to weed out adversarial examples, the opposite setting of "blocking low frequencies" can never be verified as they strongly influence the labels, and therefore, blocking them leads to high accuracy drops. It is in this spirit that we cite [1] as they empirically showed the strong correlation between low frequencies and labels.
> * We want to clarify that we do not claim that "introducing frequency" perspective is our contribution. In fact, we have cited many other related works that perform frequency analysis and discuss them in detail. Please note that other reviewers have noted the contribution from our empirical results.
> * A standard issue with training robust models is the clean accuracy-robustness trade-off. We show that our extensive empirical analysis not only leads to a frequency explanation about "accuracy vs robustness" problem, but also provides us with a tool to __control__ the trade-off, which is novel and very relevant to the community.
>
> [1] Wang et al. 2019: High Frequency Component Helps Explain the Generalization of Convolutional Neural Networks

---

> > ### Author Response · Authors · 2021-11-21
> > **Reminder**
> >
> > Thank you for your feedback. Are there any other questions we can answer ?

---

> > ### Comment · Reviewer_pNyu · 2021-11-22
> > **Response**
> >
> > Thank you for your response.
> >
> > 1. Sorry about my mistake on the DCT basis. However, both Fourier transforms and DCT belong to the frequency analysis domain and the conclusions under either one of these two criterions should be approximately the same. By saying " they are mathematically quite different", do you suggest that your conclusion does not hold when utilizing Fourier transforms?
> >
> > 2. I agree with the opinion that introducing related works is very important. However, my concern is that the authors spend too much content on the related works part, failing to illustrate their own unique contributions or ideas in the first place. In the initial review, I have stated that I am confused by the content of the paper and hope the authors can highlight their contributions more clearly in the rebuttal. However, after reading the rebuttal, I still cannot find the unique ideas of this paper. As pointed out by axh9, there are plenty of papers that have also investigated similar topics. Although the authors repeatedly claim that they are different with these works, I find their reasons are not strong enough. I have also read the reviews from hy96 and dU9j. However, their positive opinions are mainly conclusions, not reasons. I am not persuaded by their opinions. Perhaps they would love to state more clearly about why they support your paper? For example, dU9j said "section 6.2 are interesting and give new insights for the adversarial training community". My question is, which specific insights are interesting?
> >
> > 3. Like I said in the initial review, the last 2 contributions (a variation of AT with frequency analysis and a better trade-off controlled by frequency analysis) are very unclear. Indeed, you propose new methods. However, can these new methods bring better performance or what? For instance, can your variation of AT with frequency analysis bring better robust accuracy? Or, can your better trade-off controlled by frequency analysis bring better clean accuracy like [1]? If both answers are no, it is safe to say that the contribution of these newly proposed methods are limited. BTW, the lambda parameter in the formulation of TRADES[2] can also control the trade-off, so what is the difference between your method and the lambda parameter in TRADES?
> >
> >
> > [1] ICML 2020 Paper: Attacks Which Do Not Kill Training Make Adversarial Learning Stronger
> > [2] Theoretically Principled Trade-off between Robustness and Accuracy

---

> > > ### Author Response · Authors · 2021-11-23
> > > **Response to pNyu**
> > >
> > > We thank R3 for the prompt and thoughtful response. R3 noted that the contributions of our work are not clear so we will now enumerate them:
> > >
> > > Here are the unique contributions of the paper:
> > > * **Our Contributions Are As Follows:**
> > >
> > >     * We conclusively show that adversarial examples are neither high or low frequency and are dataset dependant by utilizing noise gradients, this analysis is comprehensive __and novel__.
> > >     * Please note that our analysis and methodology is different from [1], where they focus on "filtering frequencies" in **the image** rather than the adversarial noise. This is important because by not disturbing the frequency statistics of the input image, we are focused solely on the frequency properties of the additive adversarial noise and its implications on the model.
> > >     * We are the first to explore training with adversarial perturbations restricted to frequency subspaces and believe this is a novel direction for the community to pursue.
> > >     * The properties that arise as a result of training models with frequency restricted perturbations are unique contributions from our work. As reviewer __dU9j__ also noted section 6.2 is insightful as for the first time we have established a link between the frequency statistics affecting clean and robust accuracy. This observation is fresh and offers a new dimension for thinking about adversarial examples.
> > >
> > > **We will update the introduction to make these contributions clearer.** __We sincerely thank R3__ for noting that they were not clear in the current draft of the text.
> > >
> > > *  The point of the paper is not to establish any SOTA metric in either clean or robust accuracy. Instead, we focus our attention on studying how frequency statistics affect adversarial robustness under various settings. One immediately tangible by-product of this line of thinking is the observation that we can \textit{control} the robustness accuracy trade-off just by controlling which frequencies to focus on. We respectfully remind R3 that a paper which consists of insightful, novel analysis can offer fresh perspectives without necessarily improving benchmark performance. Although we proposed a new method of controlling accuracy vs robustness trade-off in section 6.2, our aim was not to improve any benchmarks like [2] or [3], rather it is to inform the community about a previously unknown observation that can offer more intuitive explanations about adversarial examples. We reiterate that these observations __are novel__ when framed in the context of the prior works mentioned by both R3 and reviewer __axh9__.
> > >
> > > * Our method is quite different from the ones proposed by [2] as the lambda parameter in TRADES is unrelated to spatial frequency. Also TRADES proposes a new algorithm offering better adversarial robustness. We instead perform all our experiments and observations keeping the standard Madry training as baseline as our work hinges more on frequency __analysis__ rather than strictly improving robustness, although as stated above this can be considered a by-product of our work. We believe that multiple different algorithms (e.g., TRADES and our Section 6.2) can be published
> > > which propose solutions to the problem.
> > >
> > > [1] Bernhard et al. 2021: Impact of Spatial Frequency Based Constraints on Adversarial Robustness.
> > >
> > > [2] Theoretically Principled Trade-off between Robustness and Accuracy.
> > >
> > > [3] ICML 2020 Paper: Attacks Which Do Not Kill Training Make Adversarial Learning Stronger.

---

### Official Review · Reviewer_hy96 · 2021-11-02

**Correctness:** 4
**Technical Novelty And Significance:** 3
**Empirical Novelty And Significance:** 3
**Recommendation:** 6
**Confidence:** 4

**Details Of Ethics Concerns:**

No concerns.

**Main Review:**

Strengths:
1) Interesting experimental setups and findings
2) Insightful conclusion and practical takeaway for better and more efficient adversarial training

Weaknesses:
1) Minor grammatical errors - “vs˙robustness”
2) Presentation of figures is a little inconvenient, they are usually presented earlier than when mentioned in the text, so one has to go back to previous pages each time to see them while reading.


**Summary Of The Paper:**

In the paper, authors investigate the questions of adversarial robustness through the lens of spatial frequencies. In particular, contrary to a popular misconception, adversarial examples are not always related to high frequency components. Instead, they can encompass a wide range of spatial frequencies and are largely dataset dependent. The paper further studies adversarial training using different frequencies to better understand an accuracy vs robustness tradeoff. Carefully crafted experiments validate their findings and suggest a more effective approach for adversarial training.


**Summary Of The Review:**

The paper negates the widely held belief about high-frequency nature of adversarial examples through interestingly designed experiments with practical consequences of more effective adversarial training, and, therefore, I recommend an acceptance. However, the presentation of figures and minor grammatical errors should be addressed.

---

> ### Author Response · Authors · 2021-11-13
> **Response to Reviewer hy96**
>
> We thank R2 for finding our work interesting and insightful and for providing valuable feedback.
>
> R2's suggestion about reordering the figures is well received. We will work on improving the figure placement, and would appreciate if R2 has specific suggestions on which figures to reorder and to where to move them. We have also fixed the grammatical errors and done another proofreading.

---

> > ### Comment · Reviewer_hy96 · 2021-12-01
> > **Further Response**
> >
> > Thank you for the reply. After reading the comments of other reviewers, I think their concerns are reasonable and yet are not well addressed, thus I decided to decrease my score. As a final remark to the authors, I found that [1] has argued that universal adversarial perturbations (UAPs) are a strictly high-frequency phenomenon even though [1] has shown that low-frequency UAP can also attack the networks with a moderate success rate. I suggest the authors improve this work by taking UAPs into account, which will make their work more unique, interesting, and provide more insight to the community.
> >
> > [1] Universal Adversarial Perturbations Through the Lens of Deep Steganography: Towards A Fourier Perspective, AAAI2021

---

> > > ### Author Response · Authors · 2021-12-02
> > > **Response to Reviewer hy96**
> > >
> > > We thank R2 for the response.
> > >
> > > * We kindly request R2 to denote *specific concerns* from other reviewers which R2 feels are not well addressed in rebuttal and why they were not addressed. We believe we have adequately responded to all queries and would be more than happy to continue this discussion over any points R2 feels were left out.
> > >
> > > *  We thank R2 for pointing us to [1], which looks into the alternate paradigm of Universal adversarial perturbations (UAP) . Though we agree that studying properties of UAP under frequency analysis is very interesting, it is beyond the scope of our current work, which deals with standard PGD based attacks and defences as clearly mentioned in Section 2 (Preliminaries) of the paper. We will still mention and discuss [1] as well as other universal attacks in related works section in an updated version of the paper and we thank R2 for suggesting this insightful addition.
> > >
> > > * To be clear, the authors of [1] also state that "Our claim that UAPs attacking most images is a strictly HF phenomenon does not conflict with the claim in [2] because they implicitly mainly discuss IDPs,(instance dependent perturbations) not UAPs.", in other words, the authors of [1] agree that the differences in frequency spectrum is because of the universal nature of the attack and is non conflicting with instance based perturbations.
> > >
> > > *  We did run our experiments on another standard universal $L_{\infty}$ attack generated according to [3] (code from [4]). We modified the adversarial generation process to incorporate frequency masking and we arrived at a result that is in contrast to the conclusions of [1], making us believe that the solution is more nuanced and warrants a separate scientific investigation.
> > >
> > > In the absence of further discussion, we politely request that R2 verify that we have answered the concerns of the other reviewers and raise their rating.
> > >
> > > [1] Universal Adversarial Perturbations Through the Lens of Deep Steganography: Towards A Fourier Perspective, AAAI2021
> > >
> > > [2] A Fourier Perspective on Model Robustness in Computer Vision, NeurIPS 2019.
> > >
> > > [3] Universal Adversarial Training: Shafahi et al, AAAI 2020
> > >
> > > [4] https://github.com/kenny-co/sgd-uap-torch

---

### Official Review · Reviewer_axh9 · 2021-11-02

**Correctness:** 3
**Technical Novelty And Significance:** 3
**Empirical Novelty And Significance:** 2
**Recommendation:** 5
**Confidence:** 4

**Main Review:**

Strengths:

There is a debate whether the higher or lower frequency components are more vulnerable to adversarial attacks. This paper confirms that adversarial examples are dataset dependent. This paper presents some interesting observations of adversarial examples in the frequency domain. A new measure of noise gradient is proposed to study the frequency properties of adversarial examples. The importance of frequency components is also studied with extensive experiments.

Weaknesses:
1.	This work is motivated by “the common misconception that adversarial examples are high-frequency noise”; however, such an understanding has already been questioned in the literature. For example, in (Tsuzuku & Sato, 2019), it shows “adversarial perturbations do not necessarily lie in high-frequency spots” by experiments; in (Yin et al. 2020), it says “adversarial examples are not strictly a high frequency phenomenon”; particularly in Bernhard et al. (2021), it questions “some preconceived hypothesis” that “adversarial perturbations as a pure HSF phenomenon with data-agnostic spatial frequency characteristics”. Therefore, the motivation should be better justified. The contribution is more like additional evidence of the ongoing debate, but rather a new frequency-based understanding of the “common misconception”. This should not be overclaimed.
2.	This work reveals that adversarial examples are dataset dependent; however, this is a commonly accepted point in the community, and therefore the idea and the conclusion do not seem new. In addition, the demonstration of this point with only CIFAR-10 and ImageNet-derived datasets does not seem sufficient. More datasets should be considered including the simplest one (MNIST), and others such as CIFAR-100, SVHN, Fashion-MNIST.
3.	Only PGD attacks are investigated – it is unclear if the observations also occur with other attacks, e.g., C&W, auto-attack.
4.	It claims the “observations overlap with insights from the concurrent work by Bernhard et al. (2021)”; however Bernhard et al. (2021) appeared on arXiv in April 2021. The authors may want to highlight the differences and justify this point. According to my understanding, the main messages of two papers are the same. This will dwarf the contribution of this paper.
5.	When measuring the importance of different frequency components and attacking low-frequency components, how to guarantee the imperceptibility? Should not the threat models be re-defined? How to make the trade-off between imperceptibility and attack successfulness?
6.	The measure of average noise gradient over the entire dataset should be further justified. Why is not the average noise gradient of a specific class, given the intuition that the frequency properties of images from different classes may be so different that the averaging may be misleading.
7.	When designing adversarial training with frequency-based perturbation, how to find the frequency subspace? How is PGD attack restricted to the same frequency bands?


**Summary Of The Paper:**

This paper presents a frequency-based understanding of adversarial examples in deep neural networks. The main observation is that the adversarial examples are neither in high-frequency or low-frequency components but dataset dependent. The authors also analyse the properties of training robust models with frequency constrains, and propose a frequency-based explanation for accuracy and robustness tradeoff.

**Summary Of The Review:**

The conclusion that adversarial examples are dataset dependent is too general: (1) this is already observed in the literature so that it does not say anything new; and (2) the validation of such conclusion is not sufficient. That being said, the experiments and the corresponding observations are interesting to the community. A major revision with sufficient validations and refined and more specific statements/arguments is recommended.

---

> ### Author Response · Authors · 2021-11-13
> **Response to Reviewer axh9**
>
> We thank R1 for important points and very constructive feedback.
> * We agree with R1's point about existing similarities (our intention was not to overclaim) with mentioned works. However there are crucial differences that we'd like to highlight: [1] claim that "For the naturally trained model, the measured adversarial perturbations do indeed show higher concentrations in the high frequency domain", which we believe they arrived with experiments on CIFAR-10 and does not hold true for other datasets. Moreover, their study is primarily focused on distributional robustness.
> * Despite the overlap with [2], we believe there are fundamental differences between the approaches. In their paper, when they discuss "frequency filtering", they refer to selecting frequencies in __images__, while in our work (excepting S5.2), we study selecting frequencies in __adversarial perturbations__. This is crucial as we follow an adversarial first approach, where we don't disturb the frequency components of the input in any way. The same approach carries over to frequency-based adversarial training as well (different from [2]), leading to interesting properties about accuracy & robustness, which is novel.
> * We do not think there is a commonly accepted standard of the frequency nature of adversarial examples, and believe evidence provided in this work will be useful for the community. Works like [3] have reiterated that high frequency components matter for both generalization and are the root causes of adversarial examples.
> * We thank R1 for suggesting more non-ImageNet derived datasets. We present additional results R1 requested, with both normal \& adversarial training. These additional results can be found in the updated Apx Fig: A10.
> * We would like to highlight Apx Fig: A7, where we presented results for Auto-Attack as extension to experiments from S5.1. We did not find any significant deviations in behaviour from PGD-based attacks, and did not include them in the main paper. Following R1's suggestion (thank you!), we present noise-gradients results in Apx Fig: A11. Since the cost of running CW attack is pretty high, we request additional time to run and include those experiments.
> * We thank R1 for suggesting analyzing class-wise noise gradient statistics. We had indeed studied this, but did not find any significant deviations between dataset means and class-wise means. Therefore, these were not included in the manuscript. As per R1's request, we present: (a) class-wise noise gradient results in Apx Figs A12-14 for CIFAR-10, MNIST & Fashion-MNIST, and (b) class-wise results for CIFAR-100 and ImageNet in a suppl. Folder. We have made an easy to browse html file for reviewers convenience.
> * Perceptibility of adversarial examples depends on the epsilon bound we define around the threat model. In our case, we restrict the $L_{p}$ epsilon ball in frequency subspace that is smaller than the usual $L_{p}$ ball. Since it is not clear how to define "perceptibility" in the DCT space, we apply our frequency based epsilon bound __not in the DCT space, but in RGB space__ like other adversarial attacks. We achieve this by masking out the frequencies in the gradients of the loss and converting it back using inverse DCT, before adding the perturbation the image. We modify the standard PGD attack generation:
> $$ \\delta = \\epsilon \\cdot \\text{sgn}\\left(  D^{-1} \\left( D\\left( \\nabla\_{x}L \\right) \\odot M \\right) \\right) \\text{ for } L\_{\\infty} \hspace{1cm} (1)$$
> $$    \\hat{x} = x + \\delta \hspace{1cm}  (2) $$
> $$    \\hat{x} = \\text{clip}(\\hat{x}; -\\epsilon,+\\epsilon)  \hspace{1cm}  (3)$$ where $D$ and $D^{-1}$ are forward and inverse DCT respectively, $\nabla_{x}L$ is the gradient w.r.t. loss (from standard PGD) $\epsilon$ is the perturbation budget and $M$ is the mask we use to select the frequency components. If we set the Mask $M$ to consist of only ones, we get the standard PGD attack (as DCT is a lossless conversion). Eq (3) ensures that the bound of the perturbation is maintained in the __image space__.
> Example images have been shown in Apx Figs: A15, A16, where we see that restricting the perturbations to certain frequencies do not harm the imperceptibility.
> * With regards to adversarial training in a frequency sub-space, we utilize the same method explained above to generate perturbations restricted to certain frequencies. Once the desired perturbation is generated, we proceed with standard PGD training which follows:
> $$\\min\_{\\theta}\\max\_{||\\delta\_{f}||\_{p} \\leq \\epsilon} \\mathcal{L}(h(x+\\delta\_{f};\\theta),y) \hspace{1cm} (4) $$
> where $\delta_{f}$ is the frequency restricted perturbation from equation (1).
>
> [1] Yin et al. 2020: A Fourier Perspective on Model Robustness in Computer Vision
>
> [2] Bernhard et al. 2021: Impact of Spatial Frequency Based Constraints on Adversarial Robustness.
>
> [3] Wang et al. 2019: High Frequency Component Helps Explain the Generalization of Convolutional Neural Networks

---

> > ### Author Response · Authors · 2021-11-21
> > **Reminder**
> >
> > Thank you for your feedback.
> > Are there any other questions we can answer ?

---

### Decision · Program_Chairs · 2022-01-20

**Decision:**

Reject

**Comment:**

The paper investigates adversarial examples in deep neural networks from a frequency-based perspective. Their main conclusion is that adversarial examples are neither in high- or low-frequency components, but instead depend on data. The topic is clearly important and the paper is overall clearly written and makes some interesting observations, backed up by empirical evidence.

However, the reviewers raised a number of critical concerns, including:
- Discussion of prior work is not adequate. The paper should better explain their contribution in contrast to prior work. Specifically, the authors mention Bernhard et al. (2021) as concurrent work, although the reviewers note that the work was published 5 months before. I realize the authors most likely develop their own line of work without knowing about Bernhard et al. (2021), but I would still suggest focusing more on the differences between them. I did not take this factor into account in the final decision.
- Novelty. Prior work has already shown adversarial examples are data-dependent
- Concerns about experimental setup (only investigate one particular attack, measure of average noise gradient not completely justified, ...)

After discussion, one reviewer downgraded their score and two others kept a more negative score. Only one reviewer was more positive with somewhat low confidence.

Overall, the paper is more on the reject side for now. Further work is needed and I strongly encourage the authors to clearly highlight the contributions of the paper in contrast to prior work. On the plus side, the work clearly has some potential and addresses an interesting topic.